# Impact Evaluation of Cyberattacks on Connected and Automated Vehicles in Mixed Traffic Flow and Its Resilient and Robust Control Strategy

**DOI:** 10.3390/s23010074

**Published:** 2022-12-21

**Authors:** Ting Wang, Meiting Tu, Hao Lyu, Ye Li, Olivier Orfila, Guojian Zou, Dominique Gruyer

**Affiliations:** 1The Key Laboratory of Road and Traffic Engineering, Ministry of Education, 4800 Cao’an Road, Shanghai 201804, China; 2College of Transportation Engineering, Tongji University, Shanghai 201804, China; 3Faculty of Maritime and Transportation, Ningbo University, 818 Feng’hua Road, Ningbo 315211, China; 4PICS-L, IFSTTAR, University Gustave Eiffel, 25 allée des Marronniers, 78000 Versailles, France

**Keywords:** connected and automated vehicles, cyberattacks, mixed traffic, car-following model, resilient and robust control strategy

## Abstract

Connected and automated vehicles (CAVs) present significant potential for improving road safety and mitigating traffic congestion for the future mobility system. However, cooperative driving vehicles are more vulnerable to cyberattacks when communicating with each other, which will introduce a new threat to the transportation system. In order to guarantee safety aspects, it is also necessary to ensure a high level of information quality for CAV. To the best of our knowledge, this is the first investigation on the impacts of cyberattacks on CAV in mixed traffic (large vehicles, medium vehicles, and small vehicles) from the perspective of vehicle dynamics. The paper aims to explore the influence of cyberattacks on the evolution of CAV mixed traffic flow and propose a resilient and robust control strategy (RRCS) to alleviate the threat of cyberattacks. First, we propose a CAV mixed traffic car-following model considering cyberattacks based on the Intelligent Driver Model (IDM). Furthermore, a RRCS for cyberattacks is developed by setting the acceleration control switch and its impacts on the mixed traffic flow are explored in different cyberattack types. Finally, sensitivity analyses are conducted in different platoon compositions, vehicle distributions, and cyberattack intensities. The results show that the proposed RRCS of cyberattacks is robust and can resist the negative threats of cyberattacks on the CAV platoon, thereby providing a theoretical basis for restoring the stability and improving the safety of the CAV.

## 1. Introduction

At present, the global innovation trend is surging, and a new round of industrial transformation is poised to take place. Internet, mobile communication, big data, artificial intelligence, and other new technologies accelerate breakthroughs and continue to evolve, promoting the rapid development of mobile Internet and automated driving technology. Connected and automated vehicles which can realize “safe, efficient, comfortable and energy-saving” driving will also emerge as the times require. CAVs are expected to improve the characteristics of traditional traffic flow from the micro vehicle level, and then provide an effective way to solve the problems of traffic congestion, traffic efficiency, and traffic pollution. Scholars have also carried out some research to demonstrate the great potential benefits of CAVs [1,2,3]. However, with the help of diverse and advanced communication technology, the “intelligent” information exchange between vehicles and the surrounding environment/world is realized all the time. Therefore, such an open-access communication environment system increases the risk of vehicles being exposed to cyberattacks, which is an urgent and critical challenge to be solved [4]. 

In order to effectively resist cyberattacks and improve traffic safety performance, scholars have conducted a lot of research on cyberattacks, which helps us understand the impact mechanism of cyberattacks on traffic flow evolution, and lays a foundation for us to design response strategies. 

In terms of efforts to reveal the impact of cyberattacks on traffic flow characteristics, Amir et al. [5] investigated the influence of mobile reactive jamming attacks on the stability of CACC platoon, and the results showed that this attack will reduce the stability of traffic flow system. Wang et al. [6] proposed an extended car-following model to describe connected traffic dynamics under cyberattacks, the results showed that the proposed model will help to avoid collision and reduce traffic congestion under the influence of cyberattacks. Li et al. [7] studied and evaluated the impact of slight cyberattacks on CAV longitudinal security through modeling and simulation. The results showed that the impact of communication location attacks is worse than that of speed attacks. In addition, the impact of cyberattacks in vehicle acceleration phase is more severe and dangerous than that in vehicle deceleration phase. Wang et al. [8] proposed a bi-bi-layer architecture composed of both a vehicle layer and a cyber layer to explore the impact of cyberattacks on CAV platoon safety and efficiency. Dong et al. [9] proposed an evaluation framework to measure the impact of cyberattacks on traffic flow performance and analyzed and studied the impact from the aspects of attack intensity, attack range, and traffic demand through numerical simulation. Khattak et al. [10] used an infrastructure-based communication platform to discuss the impact of cyberattacks on the safety and stability of connected and automated vehicle platoons under lane changes.

Furthermore, in terms of countering the adverse impact of cyberattacks on traffic flow, Zhai et al. [11] designed a new continuous feedback controller based on lattice hydrodynamic model to suppress the impact of cyberattacks, and the effectiveness of the controller in dealing with cyberattacks and reducing traffic congestion were analyzed and verified by stability analysis and numerical simulation. Noei et al. [12] proposed a traffic microsimulation tool that can simulate conventional, automated, and connected and automated vehicles in a platoon under fault, failure, and cyberattack with optimized accuracy and simulation speed to maximize throughput and without compromising safety or string stability. Lyu et al. [13] designed a communication topology safety response system (CTSRS), and further combined with the distributed model predictive control (DMPC) to ensure the stability and security of the truck platoon even if the trucks suffer cyberattacks. Cheng et al. [14] proposes a novel intelligent driving model considering cyberattacks and heterogeneous vehicles and revealed that the traffic stability and safety under cyber-attacks can be enhanced through the high proportion of cars and the information accepted from cooperative vehicles ahead. 

In addition, some effective and robust control strategies that are not targeted at CAVs also need to be further studied and are worthy of being applied to deal with CAV cyberattacks [15,16], but we will not make a further detailed summary here.

Although some studies investigated the impact of cyberattacks and put forward the corresponding strategies, to the best of our knowledge, almost no research has been done to deal with cyberattacks from the perspective of switching acceleration controller. 

To fill this gap, this paper first takes cyberattacks and different types of CAVs into account in the Intelligent Driver Model (IDM). On this basis, an acceleration control switch is designed as a robust and resilient control strategy against cyberattacks, which can help traffic flow to restore stability and enhance security. Finally, the influence of cyberattacks on the evolution of mixed traffic flow and the role of RRCS in combating cyberattacks are revealed by numerical simulations. In particular, we also carried out a sensitivity analysis of the RRCS based on different vehicle type proportions and different vehicle distribution.

The rest of the paper is organized as follows. Section 2 establishes a car-following model of CAV mixed flow under cyberattacks. Section 3 proposes the robust and resilient control strategy against cyberattacks. In Section 4, numerical simulations are carried out to reveal the impact of cyberattacks on the evolution of mixed traffic flow, and the feasibility of the RRCS is verified by comparative experiments. Finally, Section 5 gives a general conclusion about this work and some prospects for the research direction that can be considered for future developments.

## 2. Car-Following Model of CAV Mixed Traffic Flow under Cyberattacks

### 2.1. General Assumption

The schematic diagram of the research scenario is shown below (Figure 1), covering the following three assumptions:The CAV mixed traffic flow refers to different types of connected and automated vehicles, rather than the existence of non-connected and automated vehicles;Only the longitudinal car-following behavior is considered, and the lane-changing behavior and overtaking behavior are not considered;Cyberattacks may appear in every vehicle in the CAV platoon.Each vehicle only has access to its own position and velocity data and has a communication device to receive the information transmitted by the preceding vehicle. No other sensors or information sources are available (no radar, cameras, LiDAR, etc.). In this case, the communication means is seen as a remote sensor.Delays in controller switching, actuator execution, and information transmission are ignored.

### 2.2. Cyberattacks Types and Vehicle Types

According to different research needs, previous scholars have made different classifications of cyberattacks [17,18,19,20]. Considering that we only evaluate the impact of cyberattacks on traffic flow from the perspective of vehicle dynamics, referring to Wang’s [8] classification, we summarize the cyberattacks as affecting the vehicle’s position, velocity and acceleration (Table 1). Moreover, we classify vehicles into small vehicles, medium vehicles, and large vehicles according to the vehicle length. Referring to the characteristic description of different vehicle types in the existing literature [21,22], we make reasonable assumption that the larger the vehicle is, the larger the safety headway is, and the smaller the maximum acceleration, maximum deceleration, and maximum velocity are.

### 2.3. Car-Following Model of CAV Mixed Traffic Flow with Cyberattacks

Previous scholars have proposed many classical car-following models considering different actual traffic factors [23,24,25,26]. In recent years, the car-following model in the connected and automated environment has also developed rapidly [27,28,29,30,31]. The general description of the model is as follows,
(1)$$\begin{array}{l} {\dot{v}}_{n}\left(t\right)=f\left(v_{n}\left(t\right),\;s_{n}\left(t\right),\;\Delta v_{n}\left(t\right)\right)\\ s_{n}\left(t\right)=x_{n-1}-x_{n}-l_{n-1} \end{array}$$
where
${\dot{v}}_{n}\left(t\right)$ = the acceleration of vehicle n;$v_{n}\left(t\right)$ = the velocity of vehicle n;$\Delta v_{n}\left(t\right)$ = velocity difference between vehicle n and vehicle n−1;$x_{n}$ = the position of vehicle n;$l_{n-1}$ = the length of vehicle n−1; and$s_{n}\left(t\right)$ = the gap between the front of the vehicle n and the rear of the vehicle n−1.

The IDM is widely employed in car-following modeling for CAVs [32,33,34], which was proposed by Treiber et al. [23] in 2000. The main advantage of this model is to describe the acceleration and deceleration behavior and retain complex macroscopic traffic phenomenon. In addition, Wang et al. [6,8] adopted IDM as the basic vehicle dynamics model when exploring the impact of cyberattacks on CAV traffic flow. Therefore, we also refer to and continue to select IDM, and its specific expression is as follows:(2)$$\begin{array}{l} {\dot{v}}_{n}\left(t\right)=a\left[1-{\left(\frac{v_{n}\left(t\right)}{v_{0}}\right)}^{4}-{\left(\frac{s^{*}\left(v_{n}\left(t\right),\;\Delta v_{n}\left(t\right)\right)}{s_{n}\left(t\right)}\right)}^{2}\right]\\ s^{*}\left(v_{n}\left(t\right),\;\Delta v_{n}\left(t\right)\right)=s_{0}+Tv_{n}\left(t\right)+\frac{v_{n}\left(t\right)\Delta v_{n}\left(t\right)}{2\sqrt{ab}} \end{array}$$
where
a = maximum acceleration;$v_{0}$ = desired velocity;$s^{*}\left(v_{n}\left(t\right),\;\Delta v_{n}\left(t\right)\right)$ = desired headway in the current state;$s_{0}$ = minimum gap;T = safe time headway; andb = desired deceleration.

Combining the cyberattacks types and vehicle types described above, we re-characterize the parameters directly affected in the IDM, and the specific expression is as follows,
(3)$$\begin{array}{l} {\dot{v}}_{n}\left(t\right)=a_{n}^{m}\left[1-{\left(\frac{v_{n}^{m}\left(t\right)}{v_{0}^{m}}\right)}^{4}-{\left(\frac{s_{n}^{m}{}^{*}\left(v_{n}^{m}\left(t\right),\;\Delta {\overset{↔}{v}}_{n}^{m}\left(t-{\overset{↔}{\tau }}_{n}^{m}\right)\right)}{{\overset{↔}{s}}_{n}^{m}\left(t-{\overset{↔}{\tau }}_{n}^{m}\right)}\right)}^{2}\right]\\ s_{n}^{m}{}^{*}\left(v_{n}^{m}\left(t\right),\;\Delta {\overset{↔}{v}}_{n}^{m}\left(t-{\overset{↔}{\tau }}_{n}^{m}\right)\right)=s_{0}^{m}+T_{n}^{m}v_{n}^{m}\left(t\right)+\frac{v_{n}^{m}\left(t\right)\Delta {\overset{↔}{v}}_{n}^{m}\left(t-{\overset{↔}{\tau }}_{n}^{m}\right)}{2\sqrt{a_{n}^{m}b_{n}^{m}}} \end{array}$$
(4)$$m=\left\{\begin{array}{l} 1\;vehicle\;n\;is\;small\;vehicle\\ 2\;vehicle\;n\;is\;medium\;vehicle\\ 3\;vehicle\;n\;is\;large\;vehicle \end{array}\right.$$

And different types of vehicle n correspond to different parameter values. Where
$a_{n}^{m}$ = maximum acceleration;$v_{0}^{m}$ = desired velocity;$s_{0}^{m}$ = minimum gap;$T_{n}^{m}$ = safe time headway; and$b_{n}^{m}$ = desired deceleration.${\overset{↔}{\tau }}_{n}^{m}$ = the delay under cyberattacks; $\Delta {\overset{↔}{v}}_{n}^{m}\left(t-{\overset{↔}{\tau }}_{n}^{m}\right)$ = the velocity difference under cyberattacks;$s_{n}^{m}{}^{*}\left(v_{n}^{m}\left(t\right),\;\Delta {\overset{↔}{v}}_{n}^{m}\left(t-{\overset{↔}{\tau }}_{n}^{m}\right)\right)$ = desired headway in the current state under cyberattacks;${\overset{↔}{s}}_{n}^{m}\left(t-{\overset{↔}{\tau }}_{n}^{m}\right)$ = the gap under cyberattacks;


## 3. The Robust and Resilient Control Strategy (RRCS) against Cyberattacks

In order to mitigate and resist the harmful impact of cyberattacks on traffic flow, an acceleration control switch is designed as the RRCS against cyberattacks in this section. The specific control form is as follows:(5)$$\mathrm{Acceleration}\;\mathrm{controller}=\left\{\begin{array}{ll} Controller-A & if\;\frac{\Delta x_{n}\left(t\right)}{v_{n}\left(t\right)}<T_{g}\\ Controller-IDM & if\;T_{g}\leq \frac{\Delta x_{n}\left(t\right)}{v_{n}\left(t\right)}\leq T_{u}\;\\ Controller-B & if\;\frac{\Delta x_{n}\left(t\right)}{v_{n}\left(t\right)}>T_{u} \end{array}\right.$$
where

Controller−A = the control strategy in the state of “too close vehicle gap”;

Controller−IDM = the control strategy based on the Intelligent Driver Model;

Controller−B = the control strategy in the state of “too far vehicle gap”;

$T_{g}$ = vehicle time headway threshold when triggering and switching to Controller-A; and

$T_{u}$ = vehicle time headway threshold when triggering and switching to Controller-B.

The core idea of this strategy mainly has two points. The first is to keep the safe distance between vehicles under cyberattacks, and the second is to make the vehicle dynamically adjust the acceleration and gradually restore the stability of the traffic flow. When the vehicle returns to the steady-state position, its velocity shall also reach the steady-state to realize seamless switching with IDM controller. Taking Controller-A as an example, its design motivation and design steps are as follows.

First of all, we hope that the vehicles affected by the cyberattacks will return to the equilibrium position as soon as possible after implementing the strategy. At this time, the vehicle velocity is greater than the steady-state velocity, and the headway is less than the steady-state headway. Therefore, from the perspective of kinematics, the vehicle needs to decelerate first and then accelerate, resulting in the displacement difference with the steady-state, so as to achieve the established steady-state goal, in which a velocity node $v_{n}\left(t\right)=kv_{e}$ needs to be set to connect deceleration and acceleration, k is the proportional coefficient (after preliminary simulation and verification, considering the control efficiency, we set k=5/6, which can be optimized in the future). The acceleration solution process is as follows. First, the basic kinematic equation is given as follows:(6)$$\begin{array}{l} x_{0}=v_{e}t\\ v_{e}-v_{n}\left(t\right)=a_{n}\left(t\right)\cdot t\\ v_{e}^{2}-v_{n}^{2}\left(t\right)=2a_{n}\left(t\right)x \end{array}$$
where
t = time required to reach steady-state speed;x = displacement required to reach steady-state speed;$x_{0}$ = displacement of vehicle running at steady-state velocity in time t;

Construct $x-x_{0}$ , combined with Equation (6),
(7)$$2a_{n}\left(t\right)\left(x-x_{0}\right)=2a_{n}\left(t\right)x-2a_{n}\left(t\right)v_{e}t=2a_{n}\left(t\right)x-2a_{n}\left(t\right)v_{e}\cdot \frac{v_{e}-v_{n}\left(t\right)}{a_{n}\left(t\right)}$$

Moreover, the displacement difference is equal to the gap difference, that is:(8)$$x_{0}-x=s_{n}^{e}-s_{n}\left(t\right)$$

Thus, the acceleration is:(9)$$a_{n}\left(t\right)=\frac{{\left(v_{e}-v_{n}\right)}^{2}}{2\left(s_{n}^{e}-s_{n}\left(t\right)\right)}$$

Similarly, Controller-B controls the vehicle to accelerate first and then decelerate to restore the steady state, and the specific expressions of the two acceleration control strategies are as follows:(10)$$\begin{array}{l} Controller-A:\\ {\dot{v}}_{n}\left(t\right)=\left\{\begin{array}{ll} d_{e\max} & until\;v_{n}\left(t\right)=\frac{5}{6}v_{e}\;\\ \frac{{\left(v_{e}-v_{n}\right)}^{2}}{2\left(s_{n}^{e}-s_{n}\left(t\right)\right)} & when\;\frac{5}{6}v_{e}\leq v_{n}\left(t\right)\leq v_{e} \end{array}\right. \end{array}$$
(11)$$\begin{array}{l} Controller-B:\\ {\dot{v}}_{n}\left(t\right)=\left\{\begin{array}{ll} a_{c\max} & until\;v_{n}\left(t\right)=\frac{6}{5}v_{e}\\ \frac{{\left(v_{e}-v_{n}\right)}^{2}}{2\left(s_{n}^{e}-s_{n}\left(t\right)\right)} & when\;v_{e}\leq v_{n}\left(t\right)\leq \frac{6}{5}v_{e} \end{array}\right. \end{array}$$
where
$v_{e}$ = steady-state velocity;$s_{n}^{e}$ = steady-state gap;$d_{e\max}$ = maximum deceleration; and$a_{c\max}$ = maximum acceleration.

In addition, we carried out the string stability analysis of the CAV platoon, see Appendix A for details.

As shown in Figure 2, we designed an architecture with three layers to investigate the impacts of cyberattacks and RRCS of mixed CAV flow. In the first modeling building layer, we proposed a car-following model considering different cyberattack types for mixed CAV flow based on IDM. RRCS was proposed to mitigate the bad effects of cyberattacks on mixed CAV flow in the strategy construction layer. Finally, in the numerical simulation layer, we compared spatiotemporal evolution diagrams of mixed CAV flow under cyberattacks in two cases with and without RRCS. Moreover, sensitivity analyses were conducted in different platoon compositions, vehicle distributions, and cyberattack intensities. 

## 4. Numerical Simulations and Results

### 4.1. Evolution of Traffic Flow under Different Cyberattack Scenarios with or without RRCS

In order to analyze the impact of cyberattacks on the evolution of mixed traffic flow and the feasibility of RRCS application, this section selects some typical scenarios according to the previous classification of cyberattacks for numerical simulations and makes a comparative analysis of the evolution results with and without RRCS. We set up ten vehicles to form a CAV platoon. The number of large vehicles, medium vehicles, and small vehicles can be controlled according to the set proportion, and the vehicle distribution is randomly generated. The vehicle type ratio we set in this part of the simulation is 2 large vehicles, 2 medium vehicles, and 6 small vehicles. The initial velocity of all vehicles is 12 m/s, and the same type of vehicle has the same initial headway. Table 2 summarizes the parameters of three kinds of vehicle adopted from the existing research and makes appropriate adjustments [8,29,33]. It is worth noting that IDM variables are regarded as constants in this paper for the consideration of sensitivity analysis of cyberattack-related parameters in the later paper.

#### 4.1.1. Bogus Velocity Messages

Overestimate velocity

In this scenario, the fourth vehicle was attacked from t=40s to t=60s, and the velocity information transmitted to the fifth vehicle was tampered with. This velocity value was put to 1.5 times the actual velocity of the current vehicle (4th vehicle). That is to say, the rear vehicle overestimated the velocity of the front car. Figure 3a–c describes the evolution of mixed traffic flow under bogus velocity messages attack when there is no RRCS. Due to the random distribution, the leading and sixth vehicles were medium vehicles, and the seventh and eighth vehicles were large vehicles. It can be seen from Figure 3a that during t=40s to t=60s, the fifth vehicle received a false message, and it kept accelerating to approach the vehicle in front, causing potential safety hazards. In the t=60s, the cyberattack ended and the traffic flow slowly returned to the steady state. Figure 3b,c show the evolution of velocity and headway respectively, which corresponds to Figure 3a. What is depicted in Figure 3d–f is mixed traffic flow evolution diagrams under the overestimated velocity with RRCS. We can clearly find that in the case of cyberattacks, the vehicle still keeps a certain distance from the vehicle in front, which ensures the driving safety. It can also be seen from Figure 3f that the headway (platoon state with inter-distances and vehicle positioning from the leader) has been kept in a small range under cyberattack, and the platoon can still restore stability under this control strategy when attacks disappear, which proves that this strategy can play an effective role in the process of resistance overestimate velocity attack.

Underestimate velocity

In contrast to overestimating velocity, here, the velocity information of the fourth vehicle was tampered with a lower velocity (40% of the actual value). This attack was applied in the time t=40s and then transmitted to the rear vehicle, resulting in the underestimation of the velocity of the rear vehicle to the front vehicle, which lasted 20 s and ended in t=60s. The distribution of randomly generated vehicles was that the sixth and seventh vehicles were medium vehicles, and the ninth and last vehicle were large vehicles. Figure 4a shows the running track of the CAV platoon. The fifth vehicle mistakenly continued to decelerate, resulting in a large headway, which can also be directly reflected in Figure 4b,c. Comparatively speaking, the impact of this kind of cyberattack is less critical than that of overestimating velocity attack, which will not cause vehicle collision, but it will produce unnecessary vehicle spacing, which leads to the loss and waste of road resources. Figure 4d–f depicts the impact of the underestimated velocity attack on traffic flow evolution under RRCS, which can be used for comparative analysis without RRCS. It can be found that in the case of attack, the vehicle velocity still keeps a small fluctuation, and the vehicle headway is also maintained in a good state, and the stability and safety of the fleet are guaranteed. Therefore, we have arguments to explain the feasibility and superiority of this RRCS.

#### 4.1.2. Bogus Position Messages

Overestimate position

The cyberattack set here is that the position information transmitted to the rear vehicle by the third vehicle was 15 m further than the actual position, which caused the rear vehicle to overestimate the position of the front vehicle, and the attack lasted for 20 s, starting from t=40s to t=60s. Figure 5a–c depicts the evolution of the position, velocity, and headway of the CAV platoon under overestimated position message attack without RRCS. In Figure 5a,c, when the cyberattack occurs, the fourth vehicle represented by the purple line begins to approach the third vehicle, resulting in too small headway, while in Figure 5d,f, under the attack of false position information, the vehicle is also dynamically adjusted, and there will be no scene of trolley headway. Comparing the evolution diagram of vehicle velocity under the influence of overestimating position with or without the RRCS in Figure 5b,e, we find when there is no RRCS, the velocity of the fourth cheated vehicle changes suddenly when the cyberattack occurs in t=40s and returns to normal at the end of the cyberattack in t=60s. When there is RRCS, the velocity is dynamically adjusted to maintain an appropriate headway. However, there is also a disadvantage that acceleration and deceleration are more frequent, which will have an adverse impact from the perspective of passenger comfort. Overall, the RRCS can play a positive role in dealing with the cyberattack of overestimating position information.

Underestimate position

Similarly, we set up an attack of underestimated position, tampering with the position information of the third vehicle in the 40th second, so that the position information transmitted to the fourth car is 15 m closer than the actual position. Figure 6a shows the position evolution of vehicles under the influence of cyberattacks when there is no strategy. The purple line represents the fourth vehicle cheated by the false information of the vehicle in the front, which is a small car. It can be seen from Figure 6b,c that the velocity of the fourth vehicle decelerates from the steady-state velocity of 12 m/s to 8.7 m/s in the 40th second, and the headway increases from the steady-state headway to 40 m, which has a negative impact on the whole traffic flow system. Figure 6d–f is the evolution diagram of position, velocity, and headway under the RRCS. Figure 6d shows that the fourth vehicle is a medium-sized vehicle, and the wave line with minimal amplitude means the process of the vehicle resisting the cyberattack under the influence of the control strategy. In the process of the application of the RRCS, the velocity appears more frequent acceleration and deceleration fluctuations, but it can ensure that under the influence of cyberattacks, the headway can be ensured in a stable and safe range, which can be revealed from Figure 6e,f.

#### 4.1.3. Replay Messages Attack—Replay Old Acceleration

After randomly generating a platoon of three types of vehicles, we made all vehicles start to drive at a uniform constant speed of 12 m/s. At t=30s, the leading vehicle started to accelerate at an acceleration of 0.5 m/s^2^ and lasted for 10 s. Then, the leading vehicle remained at its velocity. At t=45s, the attacker recorded the acceleration message of the subject vehicle and replayed the message to it and its followers. The attack lasted 13 s and ended at t=58s. Figure 7 describes the mixed traffic flow evolution under replay acceleration attacks with/without RRCS. In Figure 7a, the purple line and the green line corresponding to the fourth vehicle and the fifth vehicle are very close to each other from t=45s to t=58s when they are attacked, so the stability is poor and the risk is high. In Figure 7b,c, the speed of the fifth car reached 19.7 m/s, and the headway was shortened to 16.2 m. After the RRCS was added, the evolution of position, velocity, and headway (vehicle position states in the platoon from the leader position) was improved, as shown in Figure 7d–f. For example, in Figure 7f, under the influence of replay acceleration attacks, the fifth vehicle does not always approach the front vehicle or even rear end, but still maintains a suitable headway, and under this acceleration control strategy, the vehicle can return to the steady velocity and steady headway, showing the superiority of the RRCS.

#### 4.1.4. Collusion Attack

Overestimate the velocity of two different vehicles simultaneously (Type I) 

The actual scenario corresponding to this part is that the attacker forges messages from multiple vehicles at the same time to mislead the subject vehicle. Firstly, the first kind of collusion attack (Type I) means that the velocity information of two different vehicles is tampered with simultaneously. The specific setting here is that the velocity messages issued by the third vehicle and the seventh vehicle from t=40s to t=60s are tampered with 1.5 times of the actual velocity, so the network information received by the fourth vehicle and the eighth vehicle are wrong. Figure 8 shows the evolution of traffic flow under the influence of collusion attacks with/without the RRCS. Compared with the close proximity of the fourth vehicle and the third vehicle, the eighth vehicle and the seventh vehicle in Figure 8a, the evolution of driving position between the vehicles in Figure 8d was better improved, which is the role of RRCS in the cyberattack stage. In Figure 8c, the fourth vehicle is a small vehicle and the eighth vehicle is a medium vehicle. The headway of each vehicle decreases from its constant-state headway to 11.5 m and 14.0 m respectively, which causes adverse interference and impact on the safety and stability of the CAV platoon. Compared with bogus information attacks, collusion attacks generate more serious traffic risks. Figure 8d–f shows the evolution of traffic flow under the influence of collusion attack when the RRCS is added, which reveals that RRCS can still play a positive role under collusion attack. However, there is also an adverse phenomenon here, that is, the frequent velocity fluctuations in the process of resisting cyberattacks, resulting in the decline of passenger comfort.

Overestimate the velocity of one vehicle and the position of another vehicle simultaneously (Type II) 

Another type of collusion attack we tested is called Type II, which means that the velocity information of one vehicle and the position information of another vehicle are tampered with simultaneously. Specifically, the velocity information of the third vehicle is tampered with 1.5 times the actual velocity and transmitted to the fourth vehicle, and the location information of the seventh vehicle is tampered with to be 17 m farther than the actual distance and transmitted to the eighth vehicle. The whole collusion attack lasted 20 s, from t=40s to t=60s. Comprehensive comparative analysis of Figure 9 shows that the application of RRCS can play the following three roles when vehicles are attacked: the first is to make fewer vehicles have a greater impact on velocity fluctuations; the second is to make the attacked vehicles keep a reasonable headway; the third is to make the CAV platoon return to a stable state faster. Therefore, this also shows that the RRCS can also deal with different forms of collusion attacks.

### 4.2. Sensitivity Analysis for RRCS

In order to study the effect of RRCS on traffic flow stabilization when vehicles are subjected to cyberattacks, we observed and analyzed the distribution of headway of all vehicles with or without RRCS over time. It is worth noting that considering the different steady-state headway of different vehicle types, it is unreasonable to directly calculate the variance of all headway at all times for analysis. Therefore, we first calculated the deviation between the headway of all vehicles at all times and their own steady-state headway, and then analyzed the variance of this deviation as an index. The specific expression is as follows,
(12)$$\begin{array}{l} \Delta x_{n}^{e}\left(t\right)=s_{n}^{e}\left(t\right)+l_{n}\left(t\right)\\ \Delta x_{n}^{dev}\left(t\right)=\Delta x_{n}\left(t\right)-\Delta x_{n}^{e}\left(t\right)\\ c_{1}=\frac{t_{1}}{\Delta t}\\ c_{2}=\frac{t_{2}}{\Delta t}\\ \Delta x_{mean}^{dev}=\frac{\sum_{k=c_{1}}^{c_{2}}\sum_{n=1}^{10}\Delta x_{n}^{dev}\left(k\right)}{10*\left(c_{2}-c_{1}\right)}\\ \mathrm{VAR}=\frac{\sum_{k=c_{1}}^{c_{2}}\sum_{n=1}^{10}{\left(\Delta x_{n}^{dev}\left(k\right)-\Delta x_{mean}^{dev}\right)}^{2}}{10*\left(c_{2}-c_{1}\right)-1} \end{array}$$
where
$\Delta x_{n}^{e}\left(t\right)$ = steady-state headway;$\Delta x_{n}^{dev}\left(t\right)$ = the deviation between actual headway and steady headway;$c_{1}$ = count start position;$c_{2}$ = count end position;$t_{1}$ = the start time of the evolution phase used to calculate the variance, $t_{1}=40s$;$t_{2}$ = the end time of the evolution phase used to calculate the variance, $t_{2}=60s$;k = number of discrete time steps;Δt = time step, 0.01 s for each step;$\Delta x_{mean}^{dev}$ = average of all deviations;VAR = variance.

#### 4.2.1. Sensitivity Analysis of Different Platoon Composition

Furthermore, due to the necessity of exploring the universality of RRCS to deal with cyberattacks in different scenarios, we carried out numerical simulation covering more generally applicable traffic flow composition scenarios, mainly including different vehicle distribution types and proportions. 

Firstly, our simulation object is still a platoon composed of 10 vehicles, then 25 different proportion combinations are randomly generated by Gaussian mixture distribution for the proportion composition of large, medium, and small vehicles, and then 4 vehicle distribution types are randomly generated for each proportion combination. Therefore, 100 different platoon composition scenarios are randomly generated here, as shown in Table 3. In addition, the seven cyberattack types and specific forms are also consistent with the cyberattacks in the simulation in Section 4.1.

In the case of 100 randomly generated platoon compositions, we calculated the variance of headway deviation value with or without RRCS in different cyberattack scenarios, and drew the results into the box plot that can directly reflect the distribution characteristics of all variances shown in Figure 9 for comparative analysis. The relevant result data are shown in Table 4 below.

Figure 10a–d shows the box plot of variance distribution of headway deviation value in 100 traffic flow scenarios with or without RRCS under bogus messages attack. The attack type in Figure 10a is overestimated velocity information. In the absence of RRCS, the upper quartile and median of variance distribution are 52.44 and 24.95, respectively, while in the presence of RRCS, the upper quartile and median decreased to 28.09 and 16.87, respectively. Looking at Figure 10b, the simulated cyberattack form is overestimated position information attack. The upper quartile and median of variance with RRCS are 19.78 and 3.48, respectively, which is also obviously lower than that without RRCS. Figure 10c represents the cyberattack scenario corresponding to underestimated velocity information. Compared with the upper quartile and median of 18.38 and 15.55 without RRCS, the presence of RRCS significantly reduces the upper quartile and median to 17.64 and 10.85, respectively. The comparison in Figure 10d describing the cyberattack scenario of underestimating position information is significant, and the upper quartile and median of variance decrease from 21.80 and 19.23 without RRCS to 3.96 and 3.59 with RRCS. In addition, the more prominent and interesting findings are as follows: first, in the cyberattack scenario with overestimated position information, in the case of 100 different traffic flow compositions randomly generated, RRCS will play a negative role in coping with cyberattack in a few cases. Second, the effect of RRCS in the attack scenario of underestimating velocity information and underestimating position information is better than the other two scenarios of overestimating vehicle information. Overall, the above four figures reflect the positive role of RRCS proposed in this paper in dealing with the cyberattack type of bogus messages, that is, RRCS can resist the damage of cyberattack to the stability of traffic flow and help the traffic flow develop in a stable direction. 

Secondly, we discuss the impact of RRCS on traffic flow stability under cyberattack scenarios of replay old acceleration information. The box diagram on the left in Figure 10e shows the variance distribution of headway deviation without RRCS, and its upper quartile and median are 28.42 and 24.76, respectively, while the box diagram on the right is obtained after RRCS plays a role when the vehicle platoon is attacked by the network, and its upper quartile and median are reduced to 25.42 and 20.13, respectively. Although the improvement effect is not very significant in terms of variance distribution, it also means that RRCS can also play a favorable role in resisting the cyberattack of replay old acceleration and promote the stability of the platoon.

The cyberattack scenarios corresponding to Figure 10f,g are two types of collusion attacks described in Section 4.1.4. In Figure 10f, the upper quartile and median of the left box graph without RRCS are 68.98 and 42.14, respectively, which are higher than 36.48 and 23.71 of the right box diagram with RRCS. The overall trend of Figure 10g is similar to that of Figure 10f. Compared with the case without RRCS, the upper quartile and median of variance distribution with RRCS are reduced from 43.80 and 36.54 to 41.57 and 33.24. The results of the above two figures confirm the feasibility of the RRCS proposed in this paper in resisting collusion attacks and improving the resilience and robustness of vehicle platoon.

#### 4.2.2. Sensitivity Analysis of Different Attack Intensity

In the above numerical simulation, the respective cyberattack intensity under different cyber-attack scenarios considered is a fixed value, such as overestimated velocity attack of 1.5 times the actual velocity and underestimated velocity attack of 0.4 times the actual velocity. In this section, we further explore the variance distribution of headway deviation under different cyberattack scenarios when the cyberattack intensity follows the linear distribution, so as to verify that the effectiveness of the RRCS proposed in this paper in resisting cyberattacks is not accidental in individual attack intensity scenarios. The cyberattack intensity of different attack types subject to linear distribution is shown in Table 5 below.

Here, we set the composition of the platoon as two large vehicles, two medium vehicles, and six small vehicles, and the vehicle distribution was randomly formed. Figure 11 is a set of variance distribution box diagrams of headway deviation obtained by setting different cyberattack intensities under different cyberattack scenarios. Table 6 shows the specific data of the box diagram corresponding to Figure 11 and some intuitive comparison data, including the improvement value and improvement proportion of the upper quartile value and median value in the box diagram.

On the premise that the multiple of overestimated velocity attack obeys the linear distribution of 1.1 to 3.5 times, Figure 11a shows the box diagram of variance distribution of headway deviation with or without RRCS. An important discovery is that the upper quartile values of the two box graphs are equal, which actually means that when the attack velocity reaches a certain multiple, RRCS will not continue to play a role, which will need further research. Nevertheless, the median with RRCS is still 22.81% lower than that without RRCS, which shows the positive role of RRCS in overestimated velocity attack to a certain extent. Figure 11b is the box diagram drawn by setting a group of linearly distributed underestimated velocity attacks. Obviously, the upper quartile of the right box graph is even smaller than the lower quartile of the left box plot, which conveys the universal effectiveness of RRCS in resisting cyberattacks under the scenario of underestimated velocity attacks. Then, we focus on Figure 11c,d. These two box charts correspond to overestimated position attack and underestimated position attack respectively. The overestimated position satisfies the linear distribution of 5 m to 45 m away from the actual position, and the underestimated position satisfies the linear distribution of 5 m to 25 m closer to the actual position. In Figure 11c, compared with the case without RRCS to resist cyberattacks, the upper quartile and median of the box chart on the right when RRCS plays a role are reduced to 5.81 and 5.03, respectively, which significantly enhances the stability. In Figure 11d, the upper quantile and median of the box graph without RRCS are 20.07 and 10.89, which are significantly higher than the indicators of the box graph with RRCS, showing that RRCS can enhance the stability of the platoon when underestimated position attack occurs.

Then, we move to the cyberattack scenario of replay old acceleration and the replay duration is determined to obey the linear distribution of 1 s to 10 s, the box diagram is shown in Figure 11e. After RRCS played a role, the upper quartile and median of the variance distribution of headway deviation decreased by 60.53% and 61.46%, respectively, which shows that RRCS can slow down the negative impact of replay old acceleration cyberattack on the platoon.

Next, we analyze the variance distribution of the headway deviation with or without RRCS when the collusion attack obeys the linear distribution. The collusion attack corresponding to Figure 11f is that the overestimated multiple of the overestimated velocity attack suffered by two vehicles follows a linear distribution of 1.1 to 3.1 times the actual velocity. It is worth noting that by comparing the left and right box graphs in Figure 11f, we can find that RRCS cannot play a positive role in the scenario of few attacks, but from the two indicators of the upper quartile and median, RRCS can resist collusive attacks (Type I) as a whole. Finally, the premise of drawing Figure 11g is to meet the collusion attack that the velocity attack multiple obeys the linear distribution of 1.1–3.1 times and the position attack information obeys the linear distribution of 5–30 m away from the actual position. By observing and comparing the Figure 11, it can be seen that the upper quartile decreased from 58.48 without RRCs to 13.87 with RRCS, and the median decreased from 39.16 to 10.61, down 76.28% and 72.91%, respectively, which significantly shows that RRCS can maintain good stability when the platoon is subjected to collusive attacks (Type II).

One-way analysis of variance (ANOVA) was carried out to further test whether there is significant difference in the variance distribution of headway deviation with or without RRCS as shown in the Table 7 below. In most scenarios, the p-values are less than 0.05, which shows that there is significant difference in the variance distribution of headway deviation with or without RRCS under the simulation premise that the cyberattack intensity follows the linear distribution. It proves that RRCS could effectively alleviate the threat brought by cyberattacks in most scenarios with different cyberattack intensities. Moreover, the *p*-values of overestimate velocity attack scenario and collusion attack type I scenario are 0.9234 and 0.7215, respectively, which indicates that there is no significant difference in the deviation distribution of headway with or without RRCS in different cyberattack intensities. RRCS could not mitigate the negative impacts of cyberattacks in some cases of these two scenarios. More complementary strategies need to be further explored in the future.

## 5. Conclusions

Under the background of possible cyberattacks in the future connected and automated vehicles environment, this paper first builds a CAV mixed traffic flow car-following model considering cyberattacks. This will help us to understand the evolution characteristics of CAV mixed traffic flow under cyberattacks. Furthermore, we design an acceleration control switcher as a robust and resilient control strategy, so that the vehicle can switch the lower layer control strategy according to the current state under the cyberattack scenario. Finally, traffic numerical simulation experiments are carried out to study the impact of cyberattacks on the evolution of CAV mixed traffic flow with or without RRCS, and to verify the feasibility of the RRCS proposed in this paper. The conclusion mainly includes the following five points:The threat of cyberattacks to CAV mixed traffic flow is significant, and the stability and security of the CAV platoon are adversely affected;Different forms of cyberattacks will cause different forms and different degrees of harmful effects. For example, vehicles will suddenly accelerate or brake, resulting in too small or too large headways between vehicles, and may even lead to vehicle collisions;Collusive attacks have the greatest adverse impact on the CAV platoon, as they involve multiple vehicle attacks;The RRCS proposed in this paper is feasible. It can not only dynamically switch the acceleration control strategy when the vehicle is under cyberattacks, so as to maintain a safe and appropriate headway, but also ensure that the CAV platoon can gradually return to a stable state after being attacked.The results of the sensitivity analyses indicates that RRCS could effectively alleviate the threat brought by cyberattacks in most scenarios with different platoon composition, vehicle distribution and most different cyberattack intensities, which shows a strong robustness.

Of course, there are still some deficiencies in this paper and the following aspects could be further explored in the future: First, there are common lane changing and overtaking behaviors in real traffic scenarios, which should be considered in the vehicle dynamics model to better describe the characteristics of traffic flow. Secondly, some parameters such as safe headway could vary rather than a fixed value, which will help to improve the universality and persuasion of the model and strategy. Thirdly, more sensor data could be considered such as LIDAR and camera. In this way, it could be possible to merge the proposed approach (RRCS) with the perception outputs and a risk assessment system. Last but not least, the effectiveness of the RRCS proposed in this paper should be verified in various car following models. It is worth mentioning that we also preliminarily confirmed the effectiveness of RRCS in PATH’s CACC car-following model [35,36], and the specific modeling and simulation results are shown in the Appendix B.

## Figures and Tables

**Figure 1 sensors-23-00074-f001:**
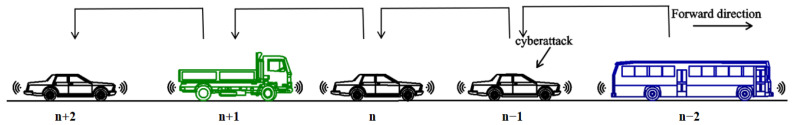
Schematic diagram of mixed CAV flow for multiple types of vehicles.

**Figure 2 sensors-23-00074-f002:**
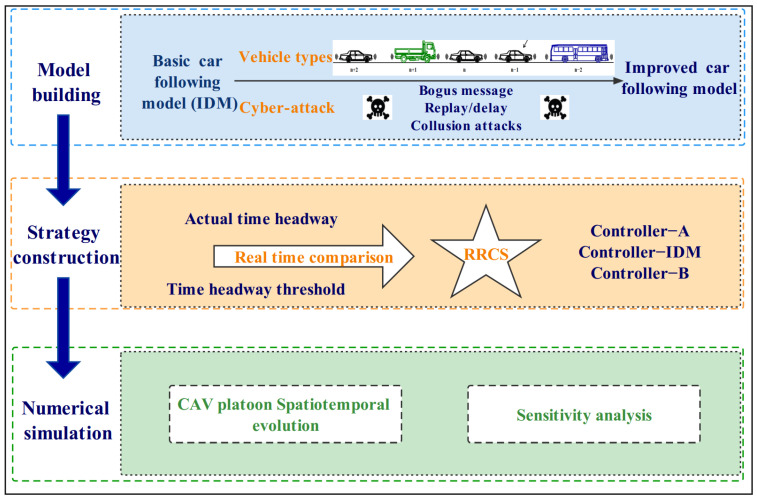
Architecture for exploring the impacts of RRCS on mixed CAV flow.

**Figure 3 sensors-23-00074-f003:**
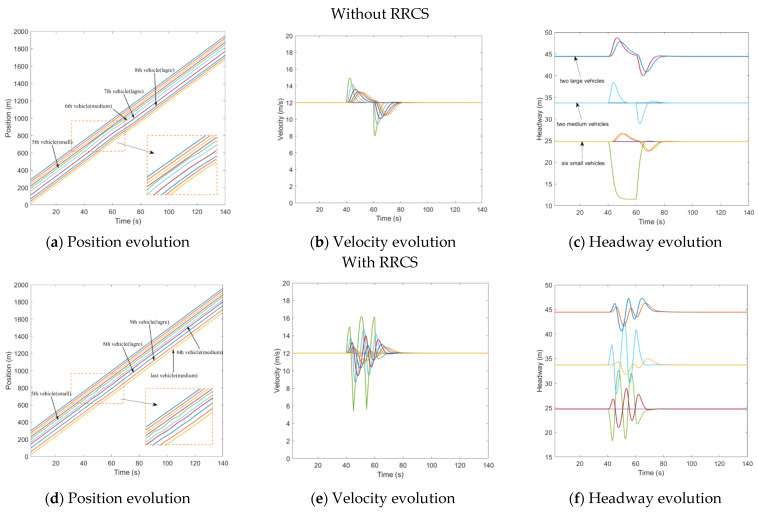
Mixed traffic flow evolution diagrams under the overestimated velocity with/without RRCS.

**Figure 4 sensors-23-00074-f004:**
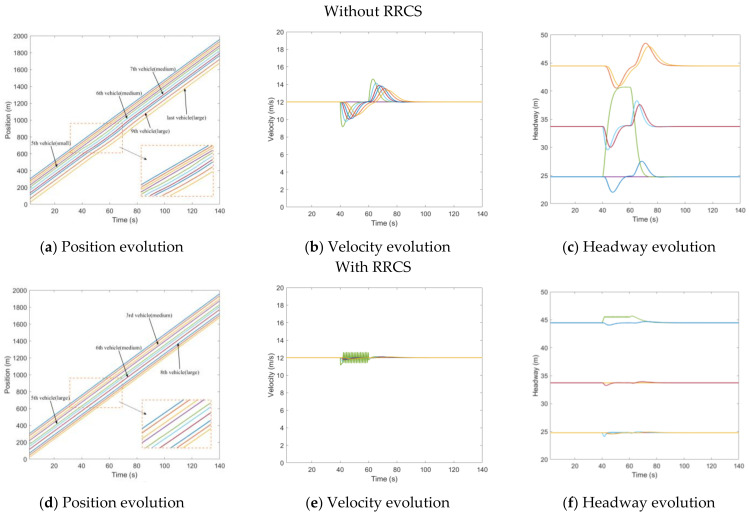
Mixed traffic flow evolution diagrams under the underestimated velocity with/without RRCS.

**Figure 5 sensors-23-00074-f005:**
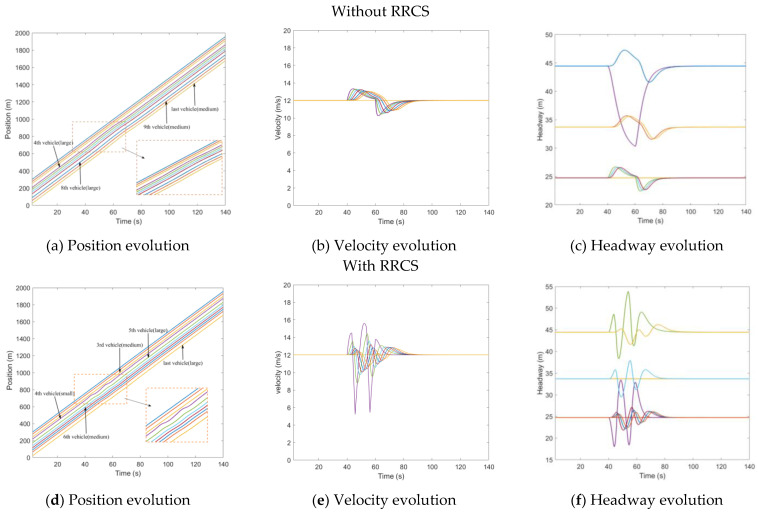
Mixed traffic flow evolution diagrams under the overestimated position with/without RRCS.

**Figure 6 sensors-23-00074-f006:**
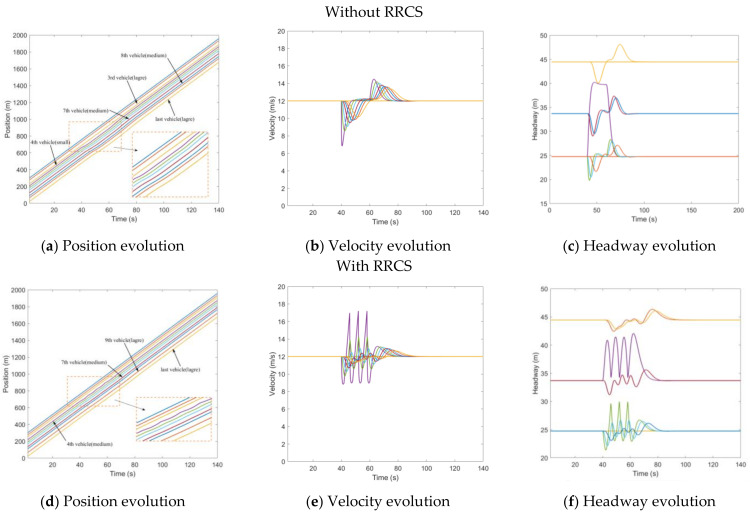
Mixed traffic flow evolution diagrams under the underestimated position with/without RRCS.

**Figure 7 sensors-23-00074-f007:**
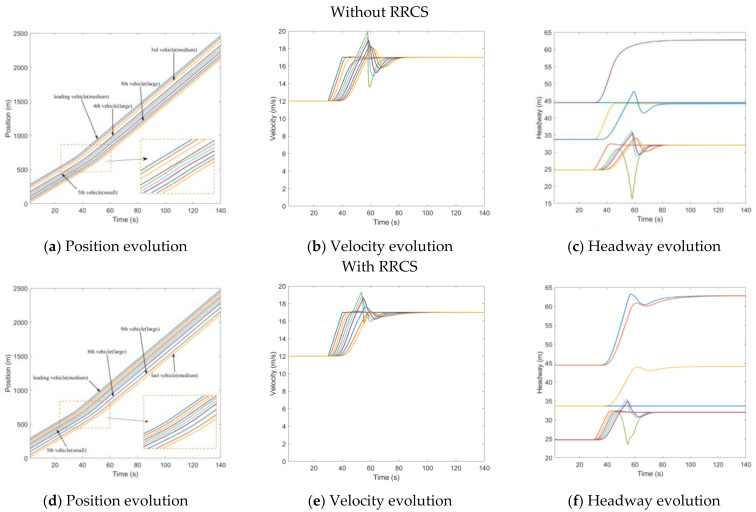
Mixed traffic flow evolution diagrams under replay acceleration attacks with/without RRCS.

**Figure 8 sensors-23-00074-f008:**
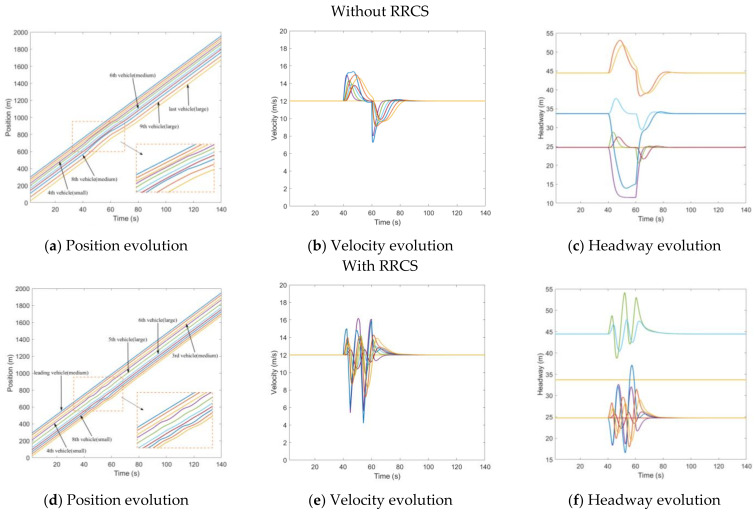
Mixed traffic flow evolution diagrams under collusion attacks (Type I) with/without RRCS.

**Figure 9 sensors-23-00074-f009:**
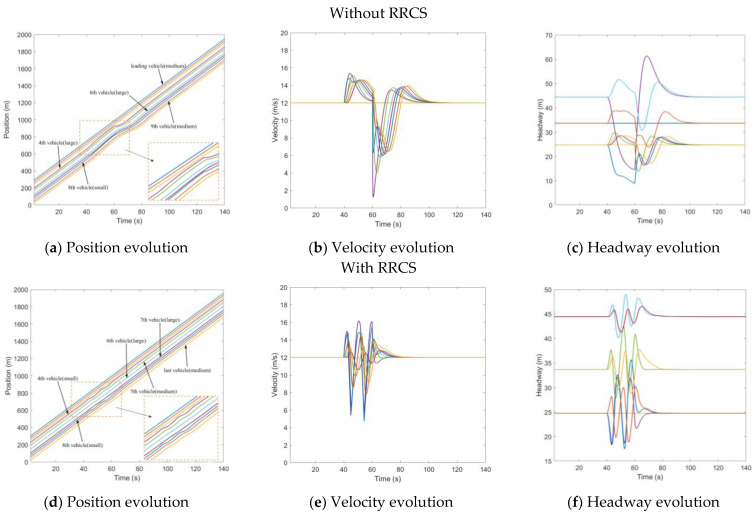
Mixed traffic flow evolution diagrams under collusion attacks (Type II) with/without RRCS.

**Figure 10 sensors-23-00074-f010:**
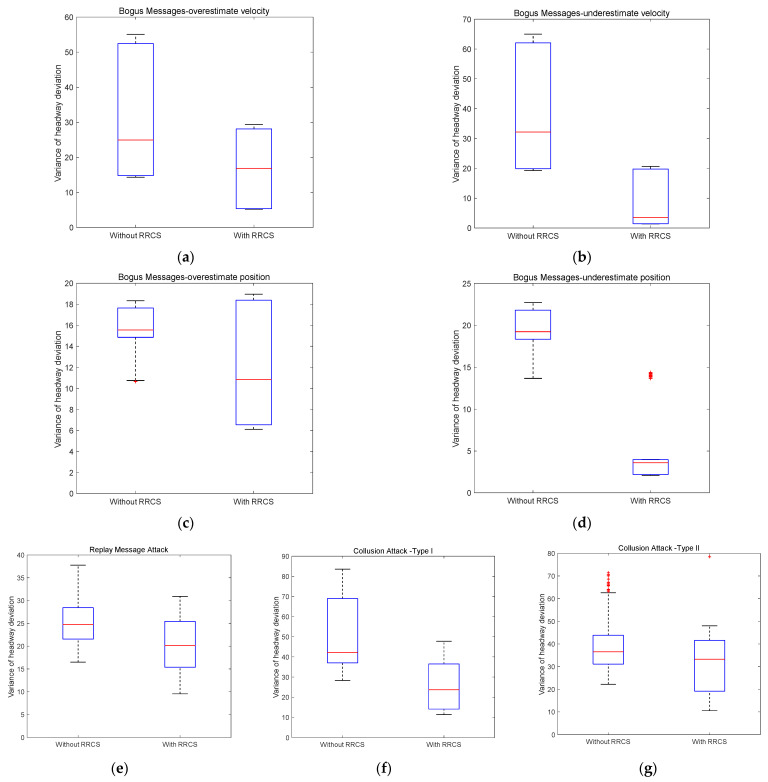
Variance distribution of headway deviation under different cyberattacks scenarios with different vehicle proportion and distribution.

**Figure 11 sensors-23-00074-f011:**
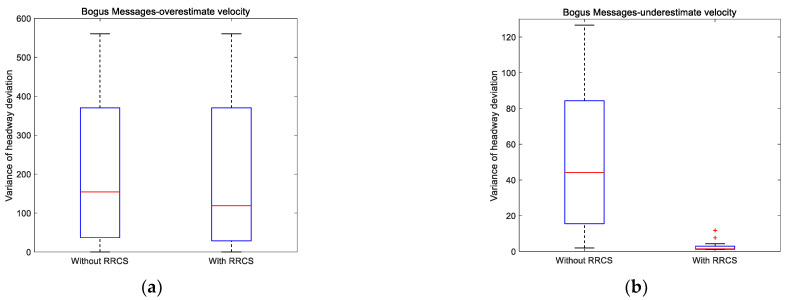
Variance distribution of headway deviation under different cyberattacks scenarios with different cyberattack intensities.

**Table 1 sensors-23-00074-t001:** Cyberattack classification.

Category	Attack Consequences
Bogus messages	Tampering with velocity information
	Tampering with location information
Replay/delay	Delayed transmission information
	Repeat transmission information
Collusion attacks	Superposition of single attacks

**Table 2 sensors-23-00074-t002:** Parameter values of different types of vehicles.

Parameter	l	$v_{0}$	a	b	T	τ	$s_{0}$	$T_{g}$	$T_{u}$	$a_{c\max}$	$d_{e\max}$
Unit	m	m/s	$m/s^{2}$	$m/s^{2}$	s	s	m	s	s	$m/s^{2}$	$m/s^{2}$
Small vehicle	5	33	2.5	3	1.3	0.1	4	1.3	3.5	2.5	4
Medium vehicle	8	27	2	2	1.6	0.15	6	1.6	3.8	2	3
Large vehicle	11	22	1.5	1	2	0.2	8	2	4	1.5	2

**Table 3 sensors-23-00074-t003:** CAV platoon composition types (100 in total).

Vehicle Proportion (Small:Medium:Large)	Vehicle Distribution
7:1:2	S-S-S-L-S-S-L-S-S-M
	S-S-S-S-L-S-L-S-S-M
	S-S-S-S-S-S-M-S-L-L
	S-M-S-S-S-S-L-S-S-L
2:6:2	M-M-M-M-M-S-M-S-L-L
	S-L-S-L-M-M-M-M-M-M
	M-M-M-S-M-M-M-L-L-S
	M-M-L-M-L-S-M-S-M-M
5:1:4	L-S-S-S-S-L-S-L-L-M
	S-L-S-S-S-L-S-L-L-M
	S-L-L-S-S-L-L-S-S-M
	L-M-S-L-L-L-S-S-S-S
. . .	. . .
6:3:1	S-M-S-S-M-L-S-M-S-S
	M-S-M-S-M-S-S-L-S-S
	S-M-M-S-S-L-S-S-S-M
	M-S-S-S-S-M-S-S-M-L

S means the small vehicle, M means the medium vehicle, L means the large vehicle.

**Table 4 sensors-23-00074-t004:** Box plot data of variance distribution of headway deviation with or without RRCS under different vehicle proportion and vehicle distribution.

Cyberattack Type	RRCS	Upper Quartile	Median	U-IV	M-IV	U-IP	M-IP
Overestimate velocity	Y	28.09	16.87	24.35	8.08	46.43%	32.38%
	N	52.44	24.95				
Underestimate velocity	Y	19.78	3.48	42.26	28.71	68.12%	89.19%
	N	62.04	32.19				
Overestimate position	Y	17.64	10.85	0.74	4.7	4.03%	30.23%
	N	18.38	15.55				
Underestimate position	Y	3.96	3.59	17.84	15.73	81.83%	81.80%
	N	21.80	19.23				
Replay old acceleration	Y	25.42	20.13	3.00	4.63	10.56%	18.70%
	N	28.42	24.76				
Collusion attack Type I	Y	36.48	23.71	32.50	18.43	47.12%	43.74%
	N	68.98	42.14				
Collusion attack Type II	Y	41.57	33.24	2.23	3.30	5.09%	9.03%
	N	43.80	36.54				

*Y* indicates the presence of RRCS, *N* indicates the absence of RRCS, *U-IV* means upper quartile improvement value, *M-IV* means median improvement value, *U-IP* means upper quartile improvement percentage, *M-IP* means median improvement percentage.

**Table 5 sensors-23-00074-t005:** Linearly distributed cyberattack intensity in different cyberattacks scenarios.

*Cyberattack Type*	*Specific Distribution of Cyberattack Intensity*
Overestimate velocity	Actual Velocity × (1.1~3.5)
Underestimate velocity	Actual Velocity × (0.1~0.9)
Overestimate position	Actual Position + (5~45)
Underestimate position	Actual Position − (5~25)
Replay old acceleration	Replay lasts for (1~10 s)
Collusion attack Type I	Actual Velocity × (1.1~3.1) and Actual Velocity × (1.1~3.1)
Collusion attack Type II	Actual Velocity × (1.1~3.1) and Actual Position + (5~30)

**Table 6 sensors-23-00074-t006:** Box plot data of variance distribution of headway deviation with or without RRCS under different cyberattack intensities.

Cyberattack Type	RRCS	Upper Quartile	Median	U-IV	M-IV	U-IP	M-IP
Overestimate velocity	Y	370.38	118.96	0	35.18	0%	22.82%
	N	370.38	154.14				
Underestimate velocity	Y	2.95	1.49	81.42	42.71	96.50%	96.63%
	N	84.37	44.20				
Overestimate position	Y	5.81	5.03	83.73	41.27	93.51%	89.14%
	N	89.54	46.30				
Underestimate position	Y	8.09	2.14	11.98	8.75	59.69%	80.35%
	N	20.07	10.89				
Replay old acceleration	Y	52.28	49.46	80.18	78.89	60.53%	61.46%
	N	132.46	128.35				
Collusion attack Type I	Y	22.44	13.65	14.42	6.85	39.12%	33.41%
	N	36.86	20.50				
Collusion attack Type II	Y	13.87	10.61	44.61	28.55	76.28%	72.91%
	N	58.48	39.16				

*Y* indicates the presence of RRCS, *N* indicates the absence of RRCS, *U-IV* means upper quartile improvement value, *M-IV* means median improvement value, *U-IP* means upper quartile improvement percentage, *M-IP* means median improvement percentage.

**Table 7 sensors-23-00074-t007:** One-way ANOVA with or without RRCS under different cyberattacks scenarios.

Scenarios	Source	SS	df	MS	F	*p*-Value
Overestimate velocity	Columns	212.1	1	212.1	0.01	0.9234
	Error	538,997.7	24	22,458.2		
	Total	539,209.8	25			
Underestimate velocity	Columns	20,369.7	1	20,369.7	24.49	2.30386 × 10^−5^
	Error	26,612.2	32	831.6		
	Total	46,981.9	33			
Overestimate position	Columns	25,993.6	1	25,993.6	27.4	5.58769 × 10^−6^
	Error	37,942.6	40	948.6		
	Total	63,936.2	41			
Underestimate position	Columns	295.68	1	295.683	4.93	0.0434
	Error	839.2	14	59.943		
	Total	1134.88	15			
Replay old acceleration	Columns	33,146.9	1	33,146.9	1065.52	8.0089 × 10^−19^
	Error	622.2	20	31.1		
	Total	33,769.1	21			
Collusion attack Type I	Columns	49.23	1	49.23	0.13	0.7215
	Error	5982.45	16	373.903		
	Total	6031.68	17			
Collusion attack Type II	Columns	8345.4	1	8345.39	21.05	2.8538 × 10^−5^
	Error	20,612.5	52	396.39		
	Total	28,957.9	53			

## Data Availability

All data generated by numerical simulation included in this study are available upon request by contact with corresponding author.

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
