# Peer review of "Impact Evaluation of Cyberattacks on Connected and Automated Vehicles in Mixed Traffic Flow and Its Resilient and Robust Control Strategy"

_sensors, 2022, doi:10.3390/s23010074_

Round 1

Reviewer 1 Report

This study investigated the impacts of different cyberattacks on heterogenous CAV flow and proposed an acceleration control switcher to ensure the safety and stability of CAV platoon under cyberattacks. Overall, the writing and significance of the paper are good. The result is comprehensive, including impact analysis, control strategy design, and sensitive analysis. However, there are some concerns and comments listed in the following:

1.      The logic in the literature review is relatively weak. The authors seem to have just stacked the previous research together but did not connect them with a narrative thread.  

2.      The paper is not the first study to investigate the impacts of cyberattacks on CAV under heterogeneous traffic flow. The authors should refer to the paper named “Cheng, R., Lyu, H., Zheng, Y., & Ge, H. (2022). Modeling and stability analysis of cyberattack effects on heterogeneous intelligent traffic flow. Physica A: Statistical Mechanics and its Applications604, 127941.” Although the previous study only considered two vehicle types in the CAV platoon, the three types should not be one of the contributions, which is very limited. Therefore, the contribution of this study should be re-illustrated.

3.      In equation (3), the authors stated that they developed the extended model for IDM. However, the model is the same as the IDM model, considering reaction time, estimation errors, and temporal anticipation. The only difference is that the traffic dynamics will be tampered by cyberattacks, so such statement is inappropriate.

4.      The authors should clearly define “stability” in the paper, which is a significant concept in the paper. There are different stabilities for CAV platoon, such as local or string stability. It seems like “stability” is achieved in the simulation, but theoretically, it will be better to prove it.

5.      The authors should provide the quantitative result of “stability,” such as the L2 norm, rather than qualitative analysis from the shape.

6.      The control scheme is an “if-else” controller, which is over-simplified. The paper does not consider the rapid speed change and the acceleration jumping. Take the Figure 3(b) and (c) as an example, the magnitude of the speed change and speed variance is intensified with RRCS. The same issues are in the Figure5(b,e) and Figure6(b,e). Extra discussion of this phenomenon is needed to ensure the model’s validity.

7.      The acceleration control schemes are a piecewise function. Do the authors assume the acceleration change can be done instantaneously? If so, it is better to illustrate such assumptions in the methodology.

8.      The setting of the information flow topology should be provided, which will help the readers understand how the cyberattacks spread in the platoon. Different IFTs have distinct impacts on CAV platoons.

9.      The legend of Figures should be plotted, such as the meaning of the lines with different colors. The X-axis and Y-axis should also be provided in the enlarged part of Figure3(a) and (d). The information could help readers to understand if the disturbance decay along the upstream of traffic flow.

10.  The unit of headway should be second. The authors misuse gap and headway. However, the gap in the figure is enormous (e.g., 25m, 35m, and 45m). Please explain how to determine the parameters.

Author Response

Thank you very much for your professional comments and constructive suggestions. The response letter is attached. Please see the attachment. 

Reviewer 2 Report

This paper is well written and it can be accepted for publication after major revision. My comments are listed as follows: 1.      The theoretical contributions should be stressed in detail in Introduction. 2.      More comparisons between different algorithms (2021-2022) with the proposed method will increase the impact of the paper. 3.      The simulation studies do not take into account the effect of noise and do not demonstrate the robustness properties of the proposed algorithm. 4.      The most recent references such as: Adaptive non-singular finite time control of nonlinear disturbed cyber-physical systems with actuator cyber-attacks and time-varying delays; Optimized Fuzzy Enhanced Robust Control Design for a Stewart Parallel Robot, should be cited in this paper. 5.      Introduction section should be detailed with recent references in literature. Advantages of the proposed algorithm upon the well-known algorithms should be stressed. 6.      The detailed block diagram of the proposed approach can be added to clarify the design procedure and structure.

Author Response

(The authors gave the same response as above.)

Round 2

Reviewer 2 Report

I have no more comments. The paper is revised well. 

Author Response

We would like to thank the reviewer. The comments helped to improve our manuscript.